# Learning Automata from Demonstrations, Examples, and Natural Language

**Marcell Vazquez-Chanlatte**[1] [*]         **Karim Elmaaroufi**[1] [*]         **Stefan Witwicki**[2]

**Matei Zaharia**[2]                                **Sanjit A. Seshia**[2]

## Abstract

Expert demonstrations have proven to be an easy way to indirectly specify complex tasks. Recent algorithms even support extracting unambiguous formal specifications, e.g. deterministic finite automata (DFA), from demonstrations. Unfortunately, these techniques are typically not sample-efficient. In this work, we introduce $L^*LM$, an algorithm for learning DFAs from both demonstrations *and* natural language. Due to the expressivity of natural language, we observe a significant improvement in the data efficiency of learning DFAs from expert demonstrations. Technically, $L^*LM$ leverages large language models to answer membership queries about the underlying task. This is then combined with recent techniques for transforming learning from demonstrations into a sequence of labeled example learning problems. In our experiments, we observe the two modalities complement each other, yielding a powerful few-shot learner.

## 1 Introduction

Large Language Models (LLMs) have emerged as a powerful tool for converting natural language expressions into structured tasks (Yang et al., 2023a; Song et al., 2023; Huang et al., 2022). Similarly, in many settings (e.g. robotics), demonstrations and labeled examples provide a complementary way to provide information about a task (Ravichandar et al., 2020). In addition, natural language has been shown to significantly reducing the number of demonstrations needed to learn to perform a task (Sontakke et al., 2023). Although impressive, these methods suffer a core limitation: they do not provide a well-defined artifact that *unambiguously* encodes the specification of the task in a manner that supports: **(i)** formal analysis and verification, and **(ii)** *composition* of tasks.

For example, we may wish to compositionally learn two task specifications independently in environments that facilitate learning them and then compose them afterwards: "dry off before recharging" or "enter water before traversing through hot regions". Similarly, due to regulatory requirements, we may wish to enforce an additional set of rules conjunctively with the learned specification, e.g., "never allow the vehicle to speed when children are present." In both cases, a desirable property of our learned task representation is that it can guarantee high-level system properties without retraining. Any need to fine-tune learned tasks with such properties undercuts the original purpose of learning generalizable task representations (Littman et al., 2017; Vazquez-Chanlatte et al., 2018).

To this end, we consider learning task specifications in the form of deterministic finite automata (DFA). The choice of DFAs as the concept class is motivated by three observations. First, DFAs offer simple and intuitive semantics that require only a cursory familiarity for formal languages. The only requirement to interpret them is a basic understanding of how to read flowcharts. As such, DFAs offer a balance between the accessibility of natural language and rigidity of formal semantics. Second, DFAs explicitly encode memory, making the identification of relevant memory needed to encode the task clear. Furthermore, they are the "simplest" family of formal languages to do so, since they are equivalent to having finite number of residual languages (Nerode congruences in the form of states) (Hopcroft & Ullman, 1979). Third, many existing formulations such as finite temporal logic and sequences of reach avoid tasks (go to location A, while avoiding B, then go to location C while avoiding D) are regular languages and thus are expressible as DFAs (Camacho et al., 2018).

---
[*]Equal Contribution. [1]: Nissan Advanced Technology Center - Silicon Valley (*marcell.chanlatte@nissan-usa.com*), [2]: UC Berkeley (*elmaaroufi@berkeley.edu*)

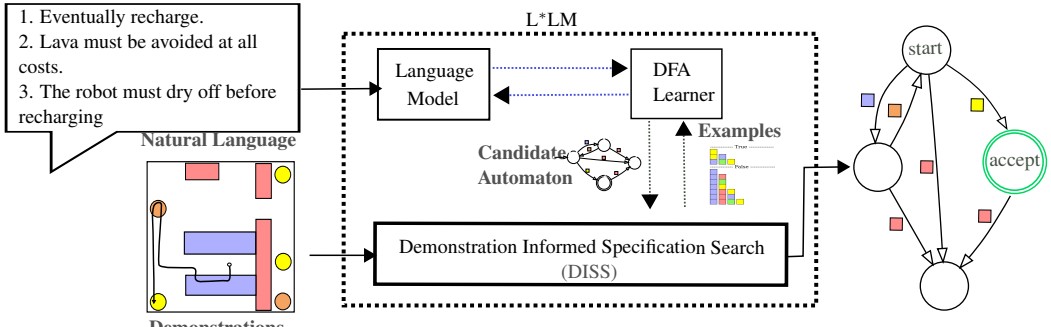

Figure 1: Our algorithm, $L^*LM$ is a multi-modal learning algorithm that builds upon the classic $L_*$ automata learning algorithm to learn automata from natural language. It also incorporates DISS to learn from expert demonstrations and self-labeled examples that minimize surprisals.

In this work, we offer three key insights that enable us to robustly learn DFAs from a mixture of demonstrations, labeled examples, and natural language. The resulting algorithm is known as $L^*LM$. The first major insight is that we design an interaction protocol built around answering simple membership queries. Here, we build on the classic automata learning literature to create an active learning algorithm that only asks the LLM membership queries, e.g., "is it ok to visit the red tile and then the blue tile?" There are two natural ways to realize this: (i) asking the LLM to synthesize code and then evaluating the code and (ii) by constraining the output of the LLM to conform to a grammar (Jones, 2023) – allowing trivial interpretation of the membership label. In this work, we focus on the latter to avoid arbitrary code execution but demonstrate in appendix B.3 that models such as GPT-4o are capable of generating membership answering programs. Importantly, as $L^*LM$ is designed to take in user inputs, avoiding arbitrary code execution removes whole classes of security vulnerabilities due to code injection.

Our second key insight is that it is important for the LLM to be able to say it is unsure when asked a membership query. We found in our experiments that the LLM would often state that it was unsure during chain-of-thought reasoning (Wei et al., 2022) followed by a hallucinated membership response. We note that this is consistent with other results in the literature (Turpin et al., 2023). The resulting DFA would then contain features that were not justified by the labeled examples or the language prompt and resulted in poor alignment with the task. A simple solution was to allow the LLM (or code generated by an LLM as in the appendix) to respond "unsure" and then have the DFA learner ask a different query. As shown in our experiments, this greatly improved performance, particularly when complemented by inferences made by analyzing the demonstrations. Note, the "unsure escape hatch" is required because the LLM is given an incomplete context to specify the task. For example, parts of the task may be omitted by the user. In our experiments *environment dynamics* are withheld from the LLM.

This leads to our third key insight: Labeled examples offer a bridge between LLM knowledge distillation and an outside verifier. In our experiments, we found that LLMs such as GPT3.5-Turbo and GPT4-Turbo failed to correctly provide membership queries for simple languages. In order to correct this, we leverage (as a blackbox) the recent Demonstration Informed Specification Search (DISS) algorithm which translates the problem of learning a DFA from an expert demonstration in a Markov Decision Process into an iterative series of DFA identification from labeled example problems (Vazquez-Chanlatte, 2022). Each problem is sent to our LLM based DFA-learner – seeded with the labeled examples as context. Because of the ability to say its unsure, the LLM is able to focus on labeling queries it is confident in due to the text prompt and leverages DISS to provide corrective feedback. By creating an interaction protocol with DISS, we fuse the LLM's natural language reasoning with dynamics-dependent analysis the LLM would otherwise be oblivious to.

**Contributions:**

1. We propose $L^*LM$, a novel algorithm for multimodal learning of deterministic finite automata from (i) natural language, (ii) labeled examples, and (iii) expert demonstrations in a Markov Decision Process.

2. A prototype implementation of $L^*LM$ written in Python and compatible with many LLMs. [1]
3. We empirically illustrate that (i) providing a natural language description of the task improves learnability of the underlying DFA, (ii) allowing the language model to respond unsure improves performance, and (iii) allowing more queries to the language model improves performance.

We emphasize that the resulting class of algorithms:
1. is guaranteed to output a valid DFA that is consistent with the input examples.
2. requires no arbitrary code evaluation.
3. only asks simple yes/no/unsure questions to the LLM.
4. supports natural language task descriptions and a-priori known examples and demonstrations.

## 2 RELATED WORK

This work lies at the intersection of a number of fields including grammatical inference, knowledge distillation from language models, and multi-modal learning. We address these connections in turn.

### 2.1 GRAMMATICAL INFERENCE AND CONCEPT LEARNING

Grammatical inference (De la Higuera, 2010) refers to the rich literature on learning a formal grammar (often an automaton (Drews & D'Antoni, 2017; Kasprzik, 2010)) from data – typically labeled examples. Specific problems include finding the smallest automata consistent with a set of positive and negative strings (De la Higuera, 2010) or learning an automaton using membership and equivalence queries (Angluin, 1987). We refer the reader to (Vaandrager, 2021) for a detailed overview of active automata learning. Notably, we leverage SAT-based DFA-identification (Ulyantsev et al., 2015; Heule & Verwer, 2010) to easily identify small DFAs that are consistent with a set of labeled examples and utilize a common technique to convert this passive learner into an active version space learner (Sverdlik & Reynolds, 1992). Our LLM to DFA extraction pipeline builds directly on these techniques. Finally, we note that the idea of learning with incomplete teachers is an evolving topic in automata learning (Moeller et al., 2023).

### 2.2 LEARNING FROM EXPERT DEMONSTRATIONS

The problem of learning objectives by observing an expert also has a rich and well developed literature dating back to early work on Inverse Optimal Control (Kalman, 1964) and more recently via Inverse Reinforcement Learning (IRL) (Ng & Russell, 2000). While powerful, traditional IRL provides no principled mechanism for composing the resulting reward artifacts and requires the relevant historical features (memory) to be a-priori known. Furthermore, it has been observed that small changes in the workspace, such as moving a goal location or perturbing transition probabilities, can change the task encoded by a fixed reward (Vazquez-Chanlatte et al., 2018; Abel et al., 2021).

To address these deficits, recent works have proposed learning Boolean task specifications, e.g. logic or automata, which admit well defined compositions, explicitly encode temporal constraints, and have workspace independent semantics (Kasenberg & Scheutz, 2017; Chou et al., 2020; Shah et al., 2018; Yoon & Sankaranarayanan, 2021; Vazquez-Chanlatte & Seshia, 2020).

Our work utilizes the Demonstration Informed Specification Search (DISS) algorithm (Vazquez-Chanlatte, 2022). DISS is a variant of maximum causal entropy IRL that recasts learning a specification (here a DFA) as a series of grammatical inference from labeled example queries. This is done by analyzing the demonstration and computing a proxy gradient over the surprisal (negative log likelihood) which suggests paths that should have their labeled changed to make the demonstrations more likely.

The key insight in our work is that this offers a bridge to our LLM extraction formalism by having the LLM fill in some examples and having DISS provide the others. From the perspective of DISS, this can be seen as indirectly restricting the concept class using a natural language prompt.

### 2.3 KNOWLEDGE EXTRACTION FROM LLMS

Attempts to extract formal specifications from language can at least be traced back to (Vadera & Meziane, 1994), if not further. More recent works have studied extracting knowledge from deep

---

[1]https://lstar-lm.github.io

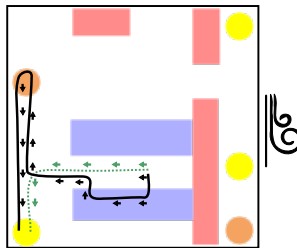 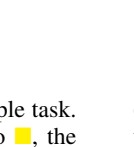 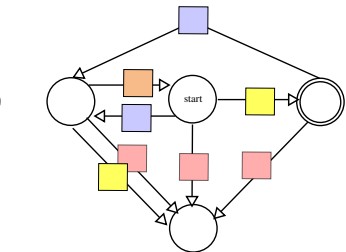

(a) Two demonstrations of an example task. While one successfully navigates to ■, the other demonstrates an implicit rule which we have omitted: dry off before recharging.

(b) DFAs with stuttering semantics. If a transition is not provided, a self-loop is assumed. Accepting states are marked with a concentric circle, and the initial state is labeled start.

Figure 2: Demonstrations of example robotic tasks and the corresponding DFA produced by $L^*LM$

learning models, e.g., extracting an automaton from a recurrent neural network using $L^*$ (Weiss et al., 2018) or using language prompts to synthesize programs (Desai et al., 2016). Several recent works have explored extracting finite state automata from large language models (Yang et al., 2023a;b).

A key difference between our work and the ones mentioned is our focus on multimodal learning from demonstrations *and* language. Further, as mentioned in the introduction, when compared against program synthesis, a key feature of the work is the restriction of arbitrary code to membership queries which avoids (i) security and analysis issues and (ii) guarantees the resulting concept is always a valid DFA, faithful to the input examples. This latter point is a significant difference from other automata extraction works, e.g., (Yang et al., 2023a), that focus on more direct extraction of the automata as a series of steps. Further, we note that the description of steps (i) encodes a policy rather than a task specification as studied in this paper (ii) is ultimately restricted to automata whereas the key ideas of our technique are applicable to arbitrary formal language learners.

Finally, we note that the fact that LLMs hallucinate on unknown queries is well known (Turpin et al., 2023), with some works going so far as to retrain the LLM to refuse answering such queries (Zhang et al., 2023). In our work, we do not retrain but instead relax our membership queries to allow the LLM to say "unsure", using constrained decoding (Tromble & Eisner, 2006; Geng et al., 2023).

## 2.4 Running Example

To ground our later discussion, we develop a running example. This running example is adapted from (Vazquez-Chanlatte, 2022). Consider an agent operating in a 2D workspace as shown in Fig 2a. The agent can attempt to move up, down, left, or right, but with probability $1/32$, wind will push the agent down, regardless of the agent's action. The agent can sense four types of tiles: red/lava (■), blue/water (■), yellow/recharging (■), and brown/drying (■).

We would like to instruct the robot to **(i)** avoid lava and **(ii)** eventually go to a recharge tile. To communicate this, we provide a variant of that natural language task description. Further, we provide a few demonstrations of the task as shown in Fig 2a. Unfortunately, in providing the description, we forget to mention one additional rule: if the robot gets wet, it needs to dry off before recharging.

An insight of our work is that the demonstrations provided imply this rule. In particular, the deviation after slipping implies that the direct path that doesn't dry off is a negative example. As we show in our experiments, neither the demonstrations nor the natural language alone is enough to consistently guess the correct DFA (shown on the right in Fig 2b.

## 3 Refresher on Automata Learning

In this section, we propose a general scheme for extracting a DFA from a large language model (LLM) that has been prompted with a general task description. We start with the definition of a DFA and a refresher on automata learning using examples and membership queries. This then sets us up for an interactive protocol with the LLM to extract the underlying DFA. Again, we start with the formal definition of a DFA and a set of labeled examples.

**Definition 3.1.** A **Deterministic Finite Automaton** (DFA) is a 5-tuple, $D = \langle Q, \Sigma, \delta, q_0, F \rangle$, where $Q$ is a finite set of **states**, $\Sigma$ is a finite alphabet, $\delta : Q \times \Sigma \to Q$ is the **transition function**, $q_0 \in Q$ is the **initial state**, and $F \subseteq Q$ are the **accepting states**. The transition function is lifted to strings, $\delta^* : Q \times \Sigma^* \to Q$. One says $D$ **accepts** $x$ if $\delta^*(q_0, x) \in F$. Denote by $L[D] \subseteq \Sigma^*$ the set of strings, i.e., **language**, accepted by $D$. Its complement, the set of **rejecting** strings, is denoted by $\overline{L[D]}$. A word, $x$, is said to **distinguish** $D_1$ and $D_2$ if it causes one to accept and the other reject, i.e., it $x$ lies in the symmetric difference of their languages, $x \in L[D_1] \ominus L[D_2]$. Finally, we will assume the DFA is endowed with a **size** (or complexity), mapping it to a positive real number. This will typically encode the number of bits or states needed to represent the DFA.

A collection of labeled examples, $X = (X_+, X_-)$ is a finite mutually exclusive set of words where $X_+$ and $X_-$ are called the **positive** examples and **negative** examples respectively. A DFA, $D$, is said to be consistent with $X$ if it accepts all positive examples and rejects all negative examples, i.e., $X_+ \subseteq L[D] \wedge X_- \subseteq \overline{L[D]}$ The **DFA identification problem** asks to find $k \in \mathbb{N}$ DFAs of minimal size that are consistent with a set of labeled examples.

The DFA identification problem is extremely underdetermined in general due to there being a countably infinite number of consistent DFA for any finite set of labeled examples (Gold, 1967). Moreover, for common size measures such a number of states, the identification problem is known to be NP-Hard (Gold, 1978). Nevertheless, many SAT-based implementations exist which, in practice, are able to efficiently solve the DFA identification problem (Heule & Verwer, 2010; Ulyantsev et al., 2016).

Notably, this all assumes a **passive** learner, i.e., one where the example set $X$ is a-priori provided. Alternatively, one can consider **active** learners that directly query for labels. Formally, one assumes that there is some unknown language $L^*$ and an oracle that can answer queries about $L^*$. The common model, referred to as the Minimally Adequate Teacher (MAT) (Angluin, 1987) assumes access to two types of queries: (i) **membership** queries, $M(x) = x \in L^*$, and (ii) **equivalence** queries, $E(D) = (l, x)$ where $l \in \{0, 1\}$ indicates if $L[D] \equiv L^*$. If $l = 0$, i.e., the candidate DFA is incorrect, then $x$ is a distinguishing string.

The classic algorithm for learning under a MAT is a also called $L^*$ (Angluin, 1987). $L^*$ is known to perform a polynomial number of membership queries and a linear number of equivalence queries. Unfortunately, in practice the equivalence queries are often not realizable, and thus are often approximated by **random sampling** or **candidate elimination**. In the former, one labels random words from a fixed distribution over words yielding a probably approximately correct (PAC) approximation of the underlying language (Angluin, 1987). In the latter, one uses DFA identification to find a set of consistent DFAs and queries distinguishing sequences. The guarantee in candidate elimination is then that size$(D)$ will be minimized. Note that that this leads to a folk algorithm for transforming any passive DFA identification algorithm into an active one. An example of the pseudo code for such an algorithm, `guess_dfa_VL`, is provided in Alg 1 (located in the appendix) where `find_minimal_dfas` refers to an arbitrary DFA identification algorithm.[2]

Finally, we observe that in many cases – as will be the case with our LLMs – the oracle may not be able to confidently provide membership queries. For example, if the task description provided does not cover the case provided, the LLM may be simply hallucinating the membership label. To address this case, we propose an **extended membership** query that answers true, false, or unsure. Further observe that Alg 1 applies in the extended setting by automatically ignoring unsure responses. This is in contrast to the L* algorithm which to our knowledge has no extension to support unsure responses. For our experiments with $L^*$ we map unsure responses to membership queries to false.

## 4 EXTRACTING DFAS FROM LLM INTERACTION

The previous section formulated automata learning in the passive setting and in the active setting. Notably, the active setting resulted in interaction protocols between the learner and an oracle that can answer membership queries. In this section, we formalize the observation that language models offer a natural approximation for a membership oracle as overviewed in Fig 3.

---

[2]The VL here stands for version space learning of which this algorithm can be seen as an instance of.

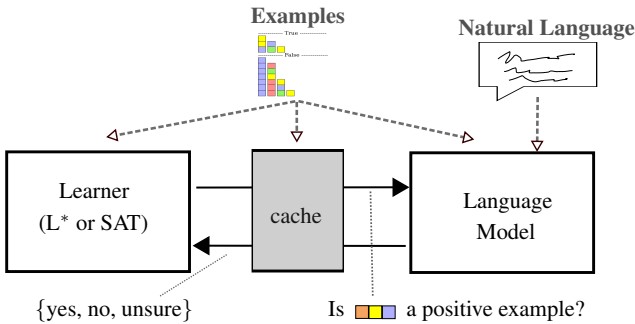

Figure 3: Visualization of the process of converting a language model into a membership oracle for DFA learning. To address the issue of hallucinations in LLMs, our work proposes an extended membership query and a cache. Ablations of our design proposals are provided in 6

Formally, a language model takes in a sequence of tokens, $x \in \Gamma^*$ called a **prompt**, and outputs another sequence of tokens $y \in \Gamma^*$ called a **response**. The response, $y$, can be viewed as a sample on the suffixes of $x$ from some underlying distribution. Techniques such as constrained decoding (Tromble & Eisner, 2006) offer the ability to further restrict $y$ to a specified formal language $L_y$ – for example using a context free grammar (Geng et al., 2023; Jones, 2023). This then guarantees that $y$ can be interpreted as an extended membership oracle. Functionally, our grammar splits the LLM's response into two parts, i.e., $y =$ work . "FINAL_ANSWER: " . answer. The work portion can be utilized to implement various prompt engineering tricks, e.g., ReAct (Yao et al., 2023) and Chain-of-Thought (Wei et al., 2022) or as seen with reasoning models (OpenAI, 2024), space for reasoning tokens. The answer part contains either yes, no, or unsure – making parsing into a membership response trivial.

During the interaction between the LLM and the DFA learner, the prompt is incrementally extended to include the responses of the previous query. The initial prompt is taken to be an arbitrary user-provided description of the task along with (i) instructions for answering the question, (ii) a request to show work, and (iii) known labeled examples. The membership query is realized by the text: "is ⟨ insert word ⟩ a positive example?"

Finally, to *guarantee* that the LLM is consistent with the provided labeled examples, we introduce a caching layer between the LLM and the DFA learner. This layer answers any known queries without consulting the LLM; furthermore, it memorizes any queries the LLM has already answered.

## 5 INCORPORATING EXPERT DEMONSTRATIONS

Next, we discuss how to introduce additional modalities. *Our key insight is that labeled examples offer a flexible late stage fusion mechanism between modalities.* For example, we leverage the Demonstration Informed Specification Search (DISS) algorithm (Vazquez-Chanlatte, 2022) which transforms the problem of learning concepts from expert demonstrations into a series of passive learning from labeled example problems.

Specifically, by an expert demonstration we mean the behavior of an agent acting in a Markov Decision Process who generates a path, $\xi$, to satisfy an objective. Here, the path is featurized into a finite alphabet and the goal is to generate a path that is accepted by some DFA $D$.

*Remark* 5.1. We observe several important differences between demonstrations and the labeled examples from the previous section. First, a demonstration of $D$ need not be accepted by $D$. That is, the labeled few-shot examples provided to the LLM must be consistent with the DFA, $D$ whereas the demonstrations need not be. Returning to our running example, this might be because (i) the robot slips and accidentally violates the task specification; or (ii) the demonstration is a prefix of the final path, perhaps being generated in real time.

The key idea of DISS is to generate counter-factual labeled examples in a manner that makes the demonstrations less surprising.

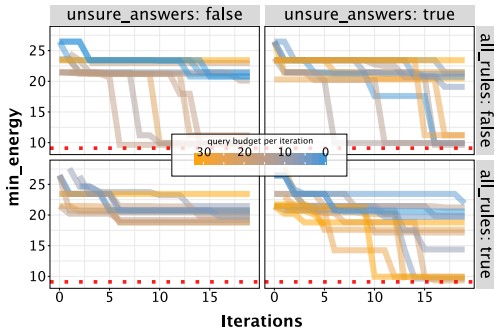 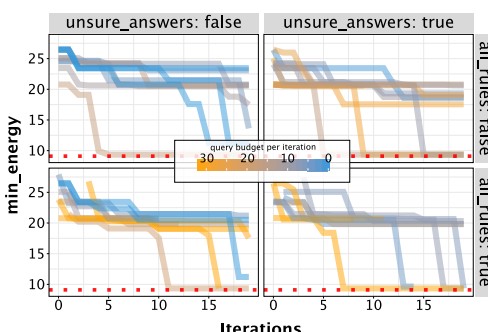

(a) Running example using $L^*$ *backend*. Ground truth energy is denoted by the red dotted line. Unlike the SAT-based backend (Fig 4b), $L^*$ struggles to learn the DFA when not provided all examples and unsure responses are not allowed.

(b) Running example using *candidate elimination variant*. In each iteration, DISS conjectures new labeled examples based on the demonstration and the conjectured DFA.

Figure 4: Comparison between $L^*$ *backend* and *candidate elimination variant* in learning DFAs.

> Our algorithm, $L^*LM\heartsuit$DISS uses DISS as a supervisory signal and as a method to incorporate expert demonstrations as an additional learning modality. This enables:
> 1. Resolving ambiguities in the natural language using demonstrations.
> 2. Indirectly restricting the concept class of DISS using a natural language prompt.

Specifically, for each iteration of DISS we run our DFA extraction algorithm from Sec 4 with a fixed query budget and split between random and candidate elimination queries. The resulting DFA is then ranked according to $U$. The processes is repeated for a fixed number of steps and the DFA with minimal energy $U$ is returned. Pseudo-code for DISS integration is provided in appendix B.

## 6 EXPERIMENTS

**Tomita Grammars** For our first experiment, we use $L^*LM$ as a membership oracle to learn DFAs representing the 7 Tomita Grammars. These simple languages are a common benchmark for studying automata learning (Weiss et al., 2018). We ask 30 membership queries for each grammar to GPT3.5-Turbo and GPT4-Turbo. We repeat each experiment for both the original $L^*$ (Angluin, 1987) learning algorithm and $L^*LM$ which allows for extended membership queries. The hallucination rates observed ($\frac{\text{\# of incorrect queries}}{\text{total \# of queries}}$) ranged from 3.33% to 100% with a median of 33.33%. In all grammars, no model was able to completely avoid hallucination, including a few experiments with GPT-4o. Full experiment details and prompts are provided in the appendix. Similar hallucination rates were also observed with Mixtral-8x7B-v0.1. These results provide two key takeaways: (i) Allowing LLMs to respond `unsure` reduces hallucination rates by up to 10%;, and (ii) despite the reduction, we are unable to eliminate hallucinations and so $L^*LM$ cannot strictly rely on an LLM to serve as an oracle. Given the need for a grounded supervisory signal, we now consider our next experiment (our running example) which provides for learning through multiple modalities.

### 6.1 ROBOTIC WORKSPACE

Given our findings in the single-mode (natural language) learning setting, we study the following research questions (RQ):
- **RQ1:** Does including a *natural language* description of the task aid in inferring the task?
- **RQ2:** Is reasoning using the demonstrations given the natural language prompt important?
- **RQ3:** Does including a *partial* description of the task aid in inferring the task?
- **RQ4:** Does allowing the LLM to state that it is unsure avoid hallucinating unhelpful components of the task?
- **RQ5:** Does $L^*$ or *dfa-identify* (our SAT-based version space DFA learning) perform better?

We adapt the experiment from (Vazquez-Chanlatte, 2022) Ch 5 which also serves as our running example to support natural language task descriptions. We use the same DFA-conditioned maximum entropy planner as (Vazquez-Chanlatte, 2022) which works on a discretized approximation of the 2D workspace. For a language model, we use Mixtral-8x7B-Instruct (Jiang et al., 2024) and

A robot is operating in a grid world and can visit four types of tiles: red, yellow, blue, green. They correspond to lava (red), recharging (yellow), water (blue), and drying (green) tiles. The robot is to visit tiles according to some set of rules. This will be recorded as a sequence of colors. Rules include:

1. The sequence must contain at least one yellow tile, i.e., eventually recharge.
2. The sequence must not contain any red tiles, i.e., lava must be avoided at all costs.
3. If blue is visited, then you must visit green *before* yellow, i.e., the robot must dry off before recharging.

A positive example must conform to all rules. Further note that repeated sequential colors can be replaced with a single instance. For example:

• [yellow,yellow,blue] => [yellow, blue]

Figure 5: Prompt used in `all_rules` treatment in experiments.

`find_minimal_dfa` is implemented using the *dfa-identify* SAT based solver (Vazquez-Chanlatte et al., 2021). The task description, which is also given as the task description to $L^*LM$ is shown in Fig 5. As in our motivating example, two demonstrations are provided illustrating the task. We instantiate several variants of $L^*LM$ varying the following factors:

1. **all_rules:** Boolean determining if the full task prompt is provided or if the third rule is replaced with $\langle$ unknown $\rangle$.
2. **allow_unsure:** Boolean determining if the response grammar includes the "unsure" output.
3. **use_L*:** Boolean determining if $L^*$ is used for the DFA-learner or `guess_dfa_VL` using SAT.
4. **query_budget**: Integer between 0 and 32 determining the number of queries allowed to the LLM per iteration.

The results are illustrated in Figures 4a and 4b for the cases use_L* = false and true, respectively. These figures plot the minimum energy, $U$, found at each step (iteration) of DISS. Each figure is broken into four quadrants varying whether all the rules are provided and whether the unsure response is included in the grammar. The color of the line indicates the number of queries allowed per DISS iteration. Finally, the red dotted line corresponds to the energy of the ground truth DFA shown in Fig 2b.

**RQ1: Natural language prompts improve performance:** Before continuing our analysis, we note a few additional details about our experiment setup.
• First, observe that with $0$ query budget (i.e. no LLM queries, we revert to the original DISS experiment as described in (Vazquez-Chanlatte, 2022). Thus, this corresponds to the performance when no natural language description is provided.
• Second, the DFA conjectured during the first iteration of DISS provides no labeled examples to the DFA learner. Thus, this corresponds to the performance of the learner with no demonstrations.
Studying Figures 4a and 4b, we clearly see that the $0$ query budget runs fail to find DFAs with equal or lower energy to the ground-truth DFA before the maximum number of DISS iterations is reached. Conversely, we see that the more both $L^*$ and the candidate elimination backend are allowed to query the LLM, the lower the final energy tends to be. Further analyzing the learned DFAs, we see that many of the runs learn the exactly correct DFA, while others learn variants that are nearly indistinguishable given the demonstration and the particular environment. We note that a similar result was observed in (Vazquez-Chanlatte, 2022) and thus, this is to be expected.

**RQ2: Analyzing the demonstrations is still required to learn the correct DFA:** Somewhat surprisingly, even in the setting where all rules are provided, we see that 5 to 15 DISS iterations are still required to learn a good DFA. As with the Tomita languages, the natural language prompt is not enough and the multi-modal nature of $L^*LM$ is indeed useful for learning correct DFAs.

**RQ3: Including more of the task description improves performance:** Comparing the top and bottom rows of Figures 4a and 4b, we see that including all of the rules has small effect with the candidate elimination backend but results in a substantial improvement for the L* backend. Notably, the candidate elimination backend seems to be able to compensate for the lack of rules using the hypothesized labeled examples from DISS.

**RQ4: Allowing unsure responses improves performance:** Comparing the left and right columns of Figures 4a and 4b, we see a clear improvement in performance when allowing the LLM to respond unsure. Again, the improvement is particularly noticeable in the L* setting. Analyzing the LLM responses, this seems to be because L* queries about words to directly determine transitions between states as opposed to directly considering what is relevant between the remaining set of consistent

small sized DFAs. This leads to queries that are largely inconsequential with a higher risk that a hallucination will lead to a larger than necessary DFA. Further, this results in DISS providing a correcting labeled example that was unnecessary given the size prior.

**RQ5: SAT-based candidate elimination outperforms $L^*$:** Finally, we observe that the SAT based candidate elimination backend which only queries distinguishing words systematically performs equal to or better than the $L^*$ backend – where the $L^*$ backend often converges just above the red line. This is particularly striking in the treatments that do not include all the rules or disallow unsure responses. As with the analysis of unsure responses, this seems to be due to $L^*$ asking queries with less utility and thus hallucinations have a larger comparative downside.

Lastly, we consider the effects of changing the output modality of the LLM. We allow the LLMs to output code instead of the context-free-grammar. Samples of the output programs are provided in the appendix. The results are similar to the bottom left graph in that the resulting programs are never able to help learn an adequate DFA. Analysis of the chains-of-thought reveal that this is because even less reasoning for the primary task is used in this setting. Instead of focusing on determining whether an example is positive or negative, the LLM quickly makes a superficial decision on the current query and instead focuses its attention on describing all it understands as a program. We also note that in this interactive setting, arbitrary code execution leaves us vulnerable to code injection attacks and mitigation strategies should be employed (e.g. sandboxes).

## 7 CONCLUSION AND DISCUSSION

In this work, we considered learning DFAs from natural language descriptions with supervision via expert demonstrations. By using the shared language of labeled examples, we showed success on a simple 2D workspace problem, even when certain rules were omitted from the description. We end with a few natural extensions of this work. Specifically, our current experiments were focused on 2D workspaces due to limitations of the current DFA conditioned maximum entropy planners used in DISS and not limitations in our proposed algorithm, $L^*LM$. Enhancing the planners to support new kinds of workspaces and larger state spaces would allow $L^*LM$ to support new types of domains ranging from IoT home automation to in-cabin assistants for cars (e.g. learn per user preferences, settings, and decisions). Finally, this work can be extended to any concept class learnable (not just DFAs) using a minimally adequate teacher (membership and approximated equivalence queries), e.g., symbolic automata (Drews & D'Antoni, 2017) and non-deterministic residual automata (Kasprzik, 2010).

## 8 ACKNOWLEDGMENTS

This work was supported in part by Provably Correct Design of Adaptive Hybrid Neuro- Symbolic Cyber Physical Systems, Defense Advanced Research Projects Agency award number FA8750-23-C-0080; by Nissan and Toyota under the iCyPhy Center; by C3DTI, by Berkeley Deep Drive, and by gifts from Accenture, AMD, Anyscale, Cisco, Google, IBM, Intel, Intesa Sanpaolo, Lambda, Mibura, Microsoft, NVIDIA, Samsung SDS, SAP, and VMware.

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

## A GUESSING DFAs AS AN ACTIVE VERSION SPACING LEARNER

---
**Algorithm 1** `guess_dfa_VL` $(\Sigma, X_+, X_-,\text{query\_budget})$

---
1: **for** $t = 1 \ldots \text{query\_budget}$ **do**
2:     $D_1, D_2 \leftarrow$ `find_minimal_dfas`$(\Sigma, X_+, X_-, 2)$
3:     word $\sim L[D_1] \ominus L[D_2]$
4:     label $\leftarrow$ M(word)
5:     **if** label = true **then**
6:       $X_+ \leftarrow$ word
7:     **else if** label = false **then**
8:       $X_- \leftarrow$ word
9: **return** `find_minimal_dfas`$(\Sigma, X_+, X_-, 1)$

---

## B PSUEDO CODE FOR L*LM

```python
# LLM wrapper prompted with task
oracle = ...
# SAT based DFA identification
find_dfa = ...
# finds word is language sym difference
distinguishing_query = ...

def guess_dfa(positive, negative)->DFA:
  # 1. Ask membership queries that
  #  distinguish remaining candidates.
  # Similar process done in
  #  equivalence query for L* backend
  for _ in range(QUERY_BUDGET):
    word = distinguishing_query(
      positive, negative, alphabet)
    label = oracle(word)
    if label is True:
      positive.append(word)
    elif label is False:
      negative.append(word)
    else: # idk case
      assert label is None

    # 2. Return minimal consistent DFA.
    return find_dfa(positive,
      negative, alphabet)

def main():
  diss = DISS()
  positive, negative = [], []
  min_nll, best = float('inf'), None
  while unsatisfied:  # DISS loop
    candidate = guess_dfa(positive,
      negative, oracle)
    # Compute counterfactual based on
    # on gradient of demonstration nll
    #  (i.e., surprisal).
    positive, negative, nll = diss.send(
      candidate)
    if nll < min_nll:
      min_nll, best = nll, candidate
  return best
```

### B.1 LLM INTERACTION EXAMPLE

**LLM interaction using direct yes/no/unsure**
Generated using ChatGPT-4o:

```
A robot is operating in a grid world and can visit four types
    of tiles: {red, yellow, blue, green}.

They correspond to lava (red), recharging (yellow), water (
    blue), and drying (green) tiles.

The robot is to visit tiles according to some set of rules.
    This will be recorded as a sequence of colors.

Rules include:

    1. The sequence must contain at least one yellow tile, i.e
    ., eventually recharge.
    2. The sequence must not contain any red tiles, i.e., lava
     must be avoided at all costs.
    3. If blue is visited, then you must visit green *before*
    yellow, i.e., the robot must dry off before recharging.

A positive example must conform to all rules.

Further note that repeated sequential colors can be replaced
    with a single
instance.

For example:
 - [yellow,yellow,blue] => [yellow, blue]
 - [red,red,blue,green,green,red] => [red,blue,green,red]
 - [blue,blue,blue] => [blue]

Additionally, by examining demonstrations of the task, we
    conjecture the following labeled examples:

POSITIVE EXAMPLES
  - [blue, green, yellow]
  - [yellow]

NEGATIVE EXAMPLES
  - [blue, red, yellow]
  - [red, yellow, green]
  - [blue, red, blue, red]
  - [blue]
  - [blue, red, green, yellow]
  - [blue, green]
  - [blue, yellow]

Please briefly answer the following questions using step-by-
    step reasoning to show your work. Do not answer any other
    question. When you arrive at a conclusion, please state it
    as FINAL_ANSWER: <yes, no>.
```

User: Is [red, red, blue] a positive example?

AI: To determine if [red, red, blue] is a positive example, we
    need to check if it conforms to all the rules.

1) The sequence must contain at least one yellow tile –
    FINAL_ANSWER: no

User: Is [yellow, blue] a positive example?

AI: To determine **if** [yellow, blue] **is** a positive example, we
    need to check **if** it conforms to **all** the rules.

1) The sequence must contain at least one yellow tile –
    FINAL_ANSWER: yes

User: Is [red, blue] a positive example?

AI: To determine **if** [red, blue] **is** a positive example, we need
    to check **if** it
conforms to **all** the rules.

1) The sequence must **not** contain **any** red tiles – FINAL_ANSWER:
    no

User: Is [yellow, yellow, green] a positive example?

AI: To determine **if** [yellow, yellow, green] **is** a positive
    example, we need to
check **if** it conforms to **all** the rules.

1) The sequence must contain at least one yellow tile –
    FINAL_ANSWER: yes

User: Is [yellow, blue, blue] a positive example?

AI: To determine **if** [yellow, blue, blue] **is** a positive example
    , we need to
check **if** it conforms to **all** the rules.

1) The sequence must **not** contain **any** red tiles – FINAL_ANSWER:
    yes

User: Is [yellow, yellow] a positive example?

AI: To determine **if** [yellow, yellow] **is** a positive example, we
    need to check **if**
it conforms to **all** the rules.

1) The sequence must contain at least one yellow tile –
    FINAL_ANSWER: yes

User: Is [yellow, red, green, yellow] a positive example?

AI: To determine **if** [yellow, red, green, yellow] **is** a positive
    example, we need to check **if** it conforms to **all** the rules.

1) The sequence must **not** contain **any** red tiles – FINAL_ANSWER:
    no

## B.2 LLM INTERACTION GENERATING CODE TO ANSWER YES/NO/UNSURE

The following was generated from ChatGPT-4o. Our prompt:

```
A robot is operating in a grid world and can visit four types
    of tiles: {red, yellow, blue, green}.

They correspond to lava (red), recharging (yellow), water (
    blue), and drying (green) tiles.

The robot is to visit tiles according to some set of rules.
    This will be recorded as a sequence of colors.

Rules include:

    1. The sequence must contain at least one yellow tile, i.e
    ., eventually recharge.
    2. The sequence must not contain any red tiles, i.e., lava
     must be avoided at all costs.

A positive example must conform to all rules.

Further note that repeated sequential colors can be replaced
    with a single instance.

For example:
 - [yellow,yellow,blue] => [yellow, blue]
 - [red,red,blue,green,green,red] => [red,blue,green,red]
 - [blue,blue,blue] => [blue]

Additionally, by examining demonstrations of the task, we
    conjecture the following labeled examples:

POSITIVE EXAMPLES
  - [blue, green, yellow]
  - [yellow]

NEGATIVE EXAMPLES
  - [blue, red, yellow]
  - [red, yellow, green]
  - [blue, red, blue, red]
  - [blue]
  - [blue, red, green, yellow]
  - [blue, green]
  - [blue, yellow]

Using step-by-step reasoning to create a python program to
    encode the above rules. Show your work. Do not answer any
    other question. The input of the python program will be a
    list of colors and the output should be "yes", "no", or "
    unsure".
```

GPT-4o's response:

```
To create a Python program that follows the rules and checks
    whether a sequence of tiles is a valid path according to
```

```
      the given conditions, we need to systematically break down
      the requirements and encode them step-by-step.

   Here's a step-by-step explanation of the logic we'll implement
      :

    Deduplicate Sequential Colors: If the same color appears
   consecutively, we will reduce them to a single instance of
   that color.

    Check for Rules:
        The sequence must contain at least one yellow tile.
        The sequence must not contain any red tiles.

    Determine Positive or Negative Example:
        If the sequence contains "red", it's an immediate "no
   ".
        If the sequence does not contain any "yellow", it's an
    immediate "no".
        If the sequence meets the above criteria but does not
   conform to the known
        patterns of positive or negative examples, it will be
   labeled as "unsure".

   Here is the Python code to achieve this:
```

```python
python

def deduplicate_sequence(sequence):
  """Remove consecutive duplicates from the sequence."""
  if not sequence:
    return []
  deduplicated = [sequence[0]]
  for color in sequence[1:]:
    if color != deduplicated[-1]:
      deduplicated.append(color)
  return deduplicated

def is_valid_sequence(sequence):
  """Check if the sequence is valid based on the given rules."""
  deduplicated = deduplicate_sequence(sequence)

  # Check for red tiles
  if "red" in deduplicated:
    return "no"

  # Check for at least one yellow tile
  if "yellow" not in deduplicated:
    return "no"

  # Compare against positive examples
  positive_examples = [
    ["blue", "green", "yellow"],
    ["yellow"]
  ]
  if deduplicated in positive_examples:
    return "yes"

  # Compare against negative examples
```

```python
negative_examples = [
  ["blue", "red", "yellow"],
  ["red", "yellow", "green"],
  ["blue", "red", "blue", "red"],
  ["blue"],
  ["blue", "red", "green", "yellow"],
  ["blue", "green"],
  ["blue", "yellow"]
]
if deduplicated in negative_examples:
  return "no"

# If it doesn't match any known patterns, return "unsure"
  return "unsure"
```

### B.3 LLM INTERACTION GENERATING CODE TO ANSWER YES/NO

The following was generated from ChatGPT-4o. Our prompt:

```
A robot is operating in a grid world and can visit four types
    of tiles: {red, yellow, blue, green}.

They correspond to lava (red), recharging (yellow), water (
    blue), and drying (green) tiles.

The robot is to visit tiles according to some set of rules.
    This will be recorded as a sequence of colors.

Rules include:

    1. The sequence must contain at least one yellow tile, i.e
    ., eventually recharge.
    2. The sequence must not contain any red tiles, i.e., lava
     must be avoided at all costs.

A positive example must conform to all rules.

Further note that repeated sequential colors can be replaced
    with a single instance.

For example:
 - [yellow,yellow,blue] => [yellow, blue]
 - [red,red,blue,green,green,red] => [red,blue,green,red]
 - [blue,blue,blue] => [blue]

Additionally, by examining demonstrations of the task, we
    conjecture the following labeled examples:

POSITIVE EXAMPLES
  - [blue, green, yellow]
  - [yellow]

NEGATIVE EXAMPLES
  - [blue, red, yellow]
  - [red, yellow, green]
  - [blue, red, blue, red]
  - [blue]
  - [blue, red, green, yellow]
  - [blue, green]
  - [blue, yellow]

Using step-by-step reasoning to create a python program to
    encode the above rules. Show your work. Do not answer any
    other question. The input of the python program will be a
    list of colors and the output should be "yes", "no".
```

ChatGPT:

```
To create a Python program that checks if a given sequence of
    tiles meets the specified rules, let's break down the steps
     needed:
```

Check for at least one yellow tile: The sequence must
contain at least one yellow tile.
Check for the absence of red tiles: The sequence must not
contain any red tiles.
Simplify the sequence: Remove consecutive repeated colors
to simplify the sequence.

Here's a step-by-step breakdown of the code:

Define the function: The function will take a **list** of
colors as **input and return** "yes" **or** "no" based on whether
the sequence meets the rules.

Simplify the sequence: Remove consecutive duplicates.

Check **for** at least one yellow tile: Ensure there **is** at
least one yellow tile **in** the simplified sequence.

Check **for** absence of red tiles: Ensure there are no red
tiles **in** the simplified sequence.

Here's the Python program that accomplishes this:

python

```python
def check_sequence(sequence):
  # Step 1: Simplify the sequence to remove consecutive duplicates
  simplified_sequence = []
  for color in sequence:
    if not simplified_sequence or simplified_sequence[-1] != color
    :
      simplified_sequence.append(color)

  # Step 2: Check for at least one yellow tile
  if 'yellow' not in simplified_sequence:
    return "no"

  # Step 3: Check for absence of red tiles
  if 'red' in simplified_sequence:
    return "no"

  # If all checks are passed
  return "yes"
```

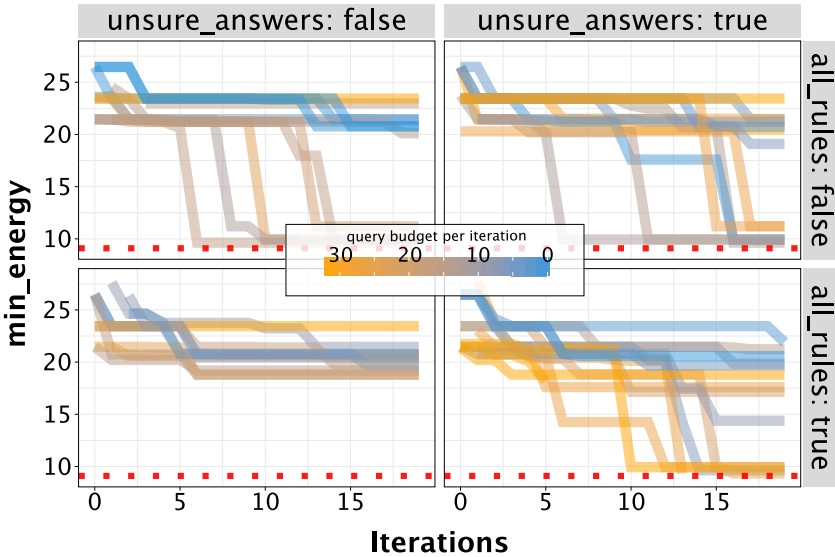

Figure 6: Running example using $L^*$ *backend.* Ground truth energy is denoted by the red dotted line. Unlike the SAT-based backend (Fig 4b), $L^*$ struggles to learn the DFA when not provided all examples and unsure responses are not allowed.

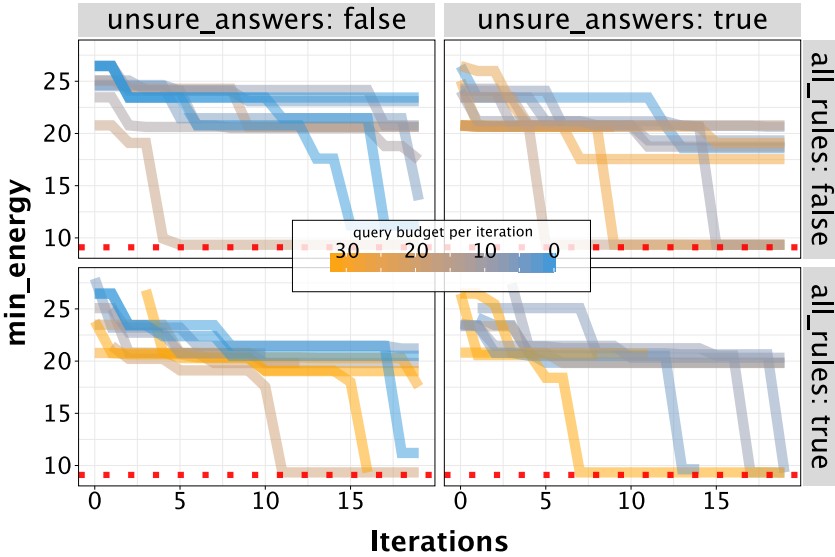

Figure 7: Running example using *candidate elimination variant.* In each iteration DISS conjectures new labeled examples based on the demonstration and the conjectured DFA.

## B.4 Comparing $L^*$ and *candidate elimination* backends

## C Tomita Grammars

### C.1 Detailed Results

The following tables summarize the results of the Tomita Grammar's experiment.

|  | Correct | Incorrect |
|---|---|---|
| Tomita 1 | 23 | 7 |
| Tomita 2 | 29 | 1 |
| Tomita 3 | 0 | 30 |
| Tomita 4 | 12 | 18 |
| Tomita 5 | 20 | 10 |
| Tomita 6 | 20 | 10 |
| Tomita 7 | 0 | 30 |

Table 1: Active learning of DFAs representing the Seven Tomita Grammars using $L^*LM$ with the original $L^*$ learning algorithm. Results are identical with both `gpt-3.5-turbo-0125` and `gpt-4-turbo`.

|  | Unsure | Correct | Incorrect |
|---|---|---|---|
| Tomita 1 | 1 | 23 | 7 |
| Tomita 2 | 1 | 29 | 1 |
| Tomita 3 | 3 | 0 | 30 |
| Tomita 4 | 1 | 12 | 18 |
| Tomita 5 | 2 | 20 | 10 |
| Tomita 6 | 2 | 20 | 10 |
| Tomita 7 | 3 | 0 | 30 |

Table 2: Active learning of DFAs representing the Seven Tomita Grammars using $L^*LM$ and the extended membership variant which includes an unsure option. Results are identical with both `gpt-3.5-turbo-0125` and `gpt-4-turbo`.

**Prompts**

The following meta-prompt was used for the treatment which allows an unsure option.

```
The following is a description of a rule for labeling a
sequence of ones and zeros as good (accepted) or
bad (rejected).

{rule}

According to the description, respond "true" if the sequence
is good and "false" if the sequence is bad. If you are unsure
or do not know the answer, respond "unsure".
Do not respond with anything else.
```

For the treatment which does not allow "unsure", simply omit the second-to-last sentence that describes unsure in the meta-prompt.

For each of the seven Tomita Grammars, we substitute the following prompts into the curly braces of the meta-prompt and query the LLM with the resultant prompt. The prompts we show here were human generated. We also repeated the same experiment with ChatGPT-4o generated prompts from providing the examples and and asking for a natural language explanation of the pattern demonstrated, and finally, we also ask for paraphrased (explained differently) prompts. We found that in any case there was no difference in the downstream $L^*LM$ results.

Tomita Grammar One:

```
The sequence should only contain the token '1'.
```

```
Seeing any other token should result in rejecting the sequence
    .

Good examples:
- 1
- 1,1
- 1,1,1
- 1,1,1,1

Bad examples:
- 0
- 1,0
- 0,1
- 1,1,0
```

Tomita Grammar Two:

```
Only accept sequences that are repetitions of 1,0.

Good examples:
- 1,0
- 1,0,1,0
- 1,0,1,0,1,0
- 1,0,1,0,1,0,1,0

Bad examples
- 1
- 1,0,1
- 0,1,0
- 1,0,1,0,0
```

Tomita Grammar Three:

```
An odd consecutive sequence of 1 should NEVER be later
followed by an odd consecutive sequence of zeros.

Good examples:
- 1,0,0
- 0,1,1,0,1,0,0
- 1,1,0,0,0
- 0,0,0,1,1,0,0,0

Bad examples
- 1,0
- 0,1,0
- 1,1,1,0,0,0
- 0,0,0,1,1,1,0,0,0
```

Tomita Grammar Four:

```
The subsequence 0,0,0 never appears, i.e., no three zeros in a
row.

Good examples:
- 1
- 1,0,0
- 0,0,1
- 1,1,0,0
```

```
Bad Examples:
- 0,0,0
- 1,0,0,0
- 0,0,0,1
- 1,1,0,0,0,1,0
```

Tomita Grammar Five:

```
There should be an even number of zeros AND an even number of
ones.

Good examples:
- 1,1
- 0,0,1,1
- 0,0,1,1,0,0
- 1,1,1,1,0,0

Bad Examples:
- 0,0,0
- 1,0,0,0
- 1,0,0,1
- 0,1,0,1,1
```

Tomita Grammar Six:

```
The difference between the number of zeros and the number of
ones is a multiple for 4.

Good examples:
- 1,0
- 0,1
- 0,1,1,1,1
- 0,1,0,1,1,1,1

Bad Examples:
- 1
- 0
- 0,1,1
- 0,1,0,1,1
```

Tomita Grammar Seven:

```
The sequence 0,1 may appear at most once in the sequence.

Good examples:
- 1
- 0,1
- 0,0,1,0
- 1,0,0

Bad examples:
- 0,1,0,1
- 1,0,1,0,1
- 0,1,1,0,1
- 0,1,0,0,1
```

