# OpenReview forum: "Learning Automata from Demonstrations, Examples, and Natural Language"
_ICLR.cc/2025/Workshop/BuildingTrust — BuildingTrust_

### Official Review · Reviewer_Zuoz · 2025-03-01

**Rating:** 7
**Confidence:** 3

**Review:**

This paper introduces L*LM, a novel algorithm for learning Deterministic Finite Automata (DFAs) from multiple modalities: natural language descriptions, labeled examples, and expert demonstrations. The authors integrate large language models (LLMs) with classic automata learning techniques and Demonstration Informed Specification Search (DISS) to create a multimodal approach to learning formal task specifications.

# Pros
- builds on well-established frameworks in automata learning, good theoretical foundation
- proposes a practical solution (allowing "unsure" responses) to LLM hallucinations in inducing automata
- uses LLMs in a relatively restricted way (answering yes/no/unsure to membership queries) to build a complex grammar

# Cons
- although the paper tests on multiple domains, they are still relatively simple. would be interesting to see if this would work on something slightly more complex

# Originality, Clarity, Significance
This is plugging together existing work like DISS, but I think the idea of using LLMs through binary membership queries is a neat one. It is more principled than directly asking the LLM to infer a grammar. I like it, I think the design is original.

---

### Official Review · Reviewer_pKyT · 2025-03-02
**Learning DFA from natural languages and demonstrations**

**Rating:** 6
**Confidence:** 2

**Review:**

The paper proposes a method to learn automata from demonstrations and natural descriptions. They integrate LLMs into either the L* or SAT methods, where the learner actively asks queries and the LLM serves as an oracle, responding to queries based on natural language descriptions with "Yes," "No," or "Unsure." To further address hallucination and incomplete description issues, they incorporate the DISS algorithm, which leverages demonstration examples to determine the DFA. Experimental results on 2D workspace problems show that this approach is effective.

---

### Decision · Program_Chairs · 2025-03-04

Accept